



# Deformation-enhanced diagenesis and bacterial proliferation in the Nankai accretionary prism

Vincent Famin[1,2], Hugues Raimbourg[3], Muriel Andreani[4], and Anne-Marie Boullier[5]

[1]Université de La Réunion, Laboratoire GéoSciences Réunion, F-97744 Saint-Denis, France.

[2]Université de Paris, Institut de Physique du Globe de Paris, CNRS, UMR 7154, F-75005 Paris, France

[3]Institut des Sciences de la Terre d'Orléans, UMR CNRS 6113, Université d'Orléans, Campus Géosciences, 1A, rue de la Férollerie, 45071 Orléans cedex 2, France.

[4]Laboratoire de Géologie de Lyon, École Normale Supérieure de Lyon et Université Claude Bernard Lyon 1, UMR 5276 CNRS, 2 rue Raphaël Dubois, 69622 Villeurbanne cedex, France.

[5]CNRS, ISTerre, Université Grenoble Alpes, F-38041 Grenoble, France.

*Correspondence to*: Vincent Famin (vincent.famin@univ-reunion.fr)

**Abstract.** Understanding diagenetic reactions in accreted sediments is critical for establishing the balance of fluid sources and sinks in accretionary prisms, which is in turn important for assessing the fluid pressure field and the ability for faults to host seismic slip. For this reason, we studied diagenetic reactions in deformation bands (shear zones and veins) within deep mud

sediments from the Nankai accretionary prism (SW Japan) drilled at site C0001 during IODP Expedition 315, by means of microscopic observation, X-ray diffraction, and major-trace element analyses. Deformation bands are not only more compacted than the host sediment, but are also enriched in framboidal pyrite, as observed under microscopy and confirmed by chalcophile element enrichments (Fe, S, Cu, As, Sb, Pb). In tandem, clays in deformation bands undergo a destabilization of smectite or illite/smectite mixed layers, and/or a slight crystallization of illite, which is matched by a correlated increase in B

and Li compared to the host sediment.

The two diagenetic reactions of sulfide precipitation and clay transformation are both explained by a combined action of sulfate-reducing and methanogen bacteria, which strongly suggests an increased activity of anaerobic microbial communities localized in deformation bands. This local bacterial proliferation was possibly enhanced by the liberation of hydrogen from strained phyllosilicates. We suggest that the proliferation of anoxic bacteria, boosted by deformation, may participate in the

pore water freshening observed at depth in accretionary prisms. Deformation-enhanced metabolic reactions may also explain the illitization observed in major faults of accretionary prisms. Care is therefore needed before interpreting illitization, and other diagenetic reactions as well, as evidence of shear heating, as these might be biogenic instead of thermogenic.

## 1 Introduction

The shallow seismicity and the stress state of convergent margins is strongly influenced by the distribution of pore fluid

pressure in the accretionary prism. For this reason, a large amount of work has been devoted to understanding the processes of fluid production, consumption or migration based on the composition of pore waters in accreted sediments. On the one hand,



this composition may be controlled by long-distance downward or upward flow, diffusing across the sediments or focused along major discontinuities such as the décollement or out of sequence thrusts (e.g. Saffer and Bekins, 1998). On the other hand, several processes of in situ fluid production or consumption can also affect the composition of pore fluids, including

metabolic reactions, organic matter cracking and mineral dehydration or alteration reactions (e.g. Carson and Screaton, 1998; Moore and Vrolijk, 1992; Raimbourg et al., 2017; Torres et al., 2015; Wallmann et al., 2006). Knowing which diagenetic reactions occur in sediments is critical for interpreting the chemistry of pore waters determined by drilling in accretionary prisms.

Diagenetic reactions are in the spotlight since the discovery of low chlorinity (i.e. lower than sea water) anomalies in sediment

pore waters from the Barbados (Gieskes et al., 1990; Moore and Vrolijk, 1992; Vrolijk et al., 1991) and Nankai accretionary prisms (Gieskes et al., 1993; Kastner et al., 1993; Underwood, 1993). Some modelling studies suggested that the transformation of smectite into illite (hereafter called illitization), assumed to be controlled essentially by temperature, might explain this pore water freshening (Brown et al., 2001; Henry and Bourlange, 2004). However, the interpretation of chlorinity is highly dependent of the porosity evolution chosen in the model, and other studies concluded that illitization alone could not account

for the observed pore water freshening (Saffer and McKiernan, 2009). This conclusion calls for an additional source of fresh fluid, either as a long-distance fluid flow or as another yet unidentified dehydration reaction. Biogenic or thermogenic processes of organic matter degradation are other diagenetic reactions capable of dramatically influencing the fluid budget of the sediment undergoing subduction, by consuming or producing water, solutes, or hydrocarbons (Pohlman et al., 2009; Raimbourg et al., 2017; Torres et al., 2015). In this category of reactions, recent studies have shown that bacterial degradation

of organic matter is able to produce large concentrations of free gas hydrocarbons (>20 $L_{gas}/L_{sediment}$), even at depths over 1000 m below sea floor (Wiersberg et al., 2015). Diagenetic reactions are also intensively studied within major faults of accretionary prisms, not only because they represent potential sinks or sources of fluids, but also because they can be used to estimate the heat generated by friction, and hence the energy dissipated by seismic ruptures (e.g. Hirono et al., 2009; Yamaguchi et al., 2011).

To improve our knowledge of diagenetic reactions in accreted sediments, we studied core samples from the NanTroSEIZE transect drilled by IODP Expedition 315 across the Nankai accretionary prism in SW Japan (Fig. 1a). Our petrographic, mineralogical, and chemical (major-and trace-elements) analyses show that small deformation bands, ubiquitously observed in sediments of the Nankai prism, host some diagenetic reactions. These diagenetic reactions may be of importance for the interpretation of pore fluid composition in deformed sediments, and for the understanding of coseismic mineral reactions in

major thrust faults within accretionary prisms.

## 2 Geologic setting

The NanTroSEIZE drilling transect in the Nankai accretionary prism is located offshore of the Kii peninsula (Fig. 1a). In this transect, the sedimentary pile of the prism may be subdivided into sediments of the Kumano forearc basin, slope or trench



sediments, and accreted sediments underneath (Fig. 1b). Accreted sediments are cut by large, landward-dipping thrust faults

called "megasplay faults". Within this transect, core drilling at sites C0001, C0002 (both drilled during IODP Expedition 315), and C0009 (Exp. 319), penetrated the Kumano forearc basin and reached the sediments of the accretionary prism underneath. Site C0004 (Exp. 316) targeted the megasplay fault located at the extreme offshore end of the Kumano basin. Sites C0006 and C0007 (Exp. 316) targeted the main frontal thrust at the edge of the accretionary prism, which was only reached at site C0007. C0008 (Exp. 316) examined the slope basin ~1 km seaward of the megasplay fault. C0011 (Exp. 319, 322 and 333) and C0012

(Exp. 322 and 333) drilled the Philippine Sea plate seaward of the deformation front.

The studied core samples come from site C0001 (Fig. 1c). At this site, sediments have been classified into two units separated by an unconformity at 207 m below sea floor (mbsf) and a ~1 Ma hiatus. Unit I represents Quaternary (0 – 2.5 Ma) slope-apron sediments. Unit II represents Middle Pliocene to Late Miocene (3.5 – 5.5 Ma) sediments of the upper accretionary prism drilled down to 456.5 mbsf. The total clay relative abundance increases in slope sediments from ~30% at the sea floor to ~60%

in the accretionary prism, in tandem with a decrease of the calcite content from ~40% to ~0% (Guo and Underwood, 2012; Kinoshita et al., 2009a). The relative abundances of quartz and plagioclase remain constant at ~20% each throughout the two units. In the clay-size fraction, smectite, illite, kaolinite, and chlorite represent ~40%, ~35%, ~5%, and ~20%, respectively. The chlorinity of pore water at C0001 decreases from 559 mM at the sea floor to 545 mM at 100 mbsf, then increases again downward. This profile indicates that a source of freshwater occurs in the first 100 mbsf of sediment. This source of freshwater

progressively vanishes at greater depth.

The studied samples all belong to the accretionary prism (Unit II) below the unconformity (Fig. 1c), which are thus essentially made of a clay-rich (~60 %) mud containing deformation bands. These deformation bands are similar to those found at other sites along the NanTroSEIZE transect and have already been described in detail in the expedition reports of IODP legs 315, 316, 319, and 333 (Ashi et al., 2008; Henry et al., 2012; Kinoshita et al., 2009a, 2009b, 2009c, 2009d, 2009e, 2009f; Saffer et

al., 2010). For this reason, the characteristic features of these microstructures are only briefly recalled here. Deformation bands include shear fractures (Fig. 2a-b) and vein structures (Fig. 2c-d) observed in macroscopic samples from split cores. Lewis et al. (2013) separated the category of shear fractures into shear zones and faults on the basis of core sample observation. Shear zones are anastomosing, dark structures up to 1 cm-thick, cutting the strata with a shear displacement of a few millimeters to centimeters (Fig. 2a). Under the microscope, shear zones appear as zones of crystallographic preferred orientations (CPO) of

phyllosilicates, which are demonstrated by their common extinction and from which the sense of shear can sometimes be reconstructed (Fig. 2b). Faults also display a mm-to-cm scale shear displacement and phyllosilicate CPOs, but are much thinner (less than 1 mm) and do not always show a visible darkening. Faults spatially evolve into shear zones or the reverse, or branch onto shear zones. This study hereafter focuses on shear zones due to their larger thickness than faults. Vein structures show the characteristic features of "ghost veins" described in soft mud sediments of continental margins worldwide (Brothers et al.,

1996; Kemp, 1990; Lindsley-Griffin et al., 1990; Ohsumi and Ogawa, 2008). They appear as 1 – 10 cm long, ~50 μm thick, dark curviplanar seams occurring in anastomosing clusters, and cutting the bedding stratification at high angle (Fig. 2c). These anastomosing clusters form comb-like, 5 – 10 cm thick arrays subparallel to the bedding planes. In thin section, veins also





display a phyllosilicate CPO parallel to the vein walls, as do shear fractures, but no or little shear displacement (Fig. 2d). Such vein arrays have been interpreted as dewatering structures occurring during the passage of earthquake waves in the soft

sediment (Brothers et al., 1996; Hanamura and Ogawa, 1993), or more recently as due to shear waves associated with density or debris flows, landslides or faulting (Ohsumi and Ogawa, 2008).

Beside the phyllosilicate CPO, the main textural difference between deformation bands and their host mudrock matrix is a pore space reduction (evidenced by X-Ray tomography scanner and field-emission secondary electron microscopy) indicating compaction, and sometimes a subtle grain size reduction (Milliken and Reed, 2010; Ujiie et al., 2004). Deformation bands are

found in all the sedimentary units of the Nankai accretionary prism. They are, however, scarce in sediments of the Kumano forearc basin or in slope sediments ($\leq$ 10 occurrences per 10 m of core, Fig. 1c), and much more abundant in accreted sediments (locally $\gg$ 30 occurrences per 10 m of core). This indicates that the majority of the deformation structures formed in the accretionary prism before the deposition of slope-apron sediments, i.e. before 2.5 Ma.

## 3 Methods

A summary of all the analyses performed on core samples from site C0001 is provided in Table 1. In a first step, standard 30 μm-thick polished thin sections were cut from core samples containing deformation bands. Petrological observations were carried out on all the samples with optical microscopy, and on four samples with a Secondary Electron Microscope (SEM) at Paris VI University. The modal proportions of opaque minerals in deformation bands and host rocks were estimated from the analysis of optical microscopic images in reflected light using the ImageJ software.

One shear zone sample (4R-3, 73-76) was found to be large enough to be analysed by X-ray diffraction (XRD), in order to compare the nature of its clay fraction relative to that of the host matrix. To do so, powders of material were collected by scrapping the shear zone structure and the host matrix, in the rock slab previously used for the preparation of the thin section. The powders were decarbonated and aqueous suspensions were prepared in a solution of 0.5 M NaCl in order to saturate clay minerals with Na. Oriented slides were prepared by drying at room temperature aqueous suspensions on monocrystalline

silicon slides to obtain an air-dried (AD) preparation. Ethylene-glycol (EG) solvation of the samples was achieved by exposing them to EG vapour at 70°C for a minimum of 12 hours. XRD patterns were recorded on the AD and EG preparations using a Bruker D8 diffractometer equipped with an MHG Messtechnik humidity controller coupled to an Anton Paar CHC+ chamber. Intensities were measured with a SolXE Si(Li) solid-state detector (Baltic Scientific Instruments) for 10 s per 0.04° 2θ step over the 2−50° 2θ Cu Kα angular range. Divergence slit, the two Soller slits, the antiscatter, and resolution slits were 0.3°,

2.3°, 0.3°, and 0.1°, respectively. Samples were kept at 23 °C and a constant 40% relative humidity (RH) in the CHC+ chamber during the whole data collection. RH was continuously monitored with a hygrometer (uncertainty of ∼2% RH) located close to the sample.

Major- and minor-element analyses were carried out on some of the polished thin sections. Element maps were performed on two samples with a X-ray fluorescence (XRF) spectrometer at Joseph Fourier University in Grenoble (France) to map major





elements on large sample surfaces, and with the SEM to map major and minor elements on smaller surfaces. On five shear
zone and vein samples, quantitative analyses of eight elements were then performed using a CAMECA SX100 electron probe
micro analyzer (EPMA) at Paris VI University. The EPMA was tuned at 15 kV and 10 nA, with a beam focused at 3 μm, a 5
s counting time for Na (measured first), Si, and K, and a 10 s counting time for other elements. The CAMECA set of standards
(synthetic and natural minerals or oxides) was used for calibration. The correction methods of Bence and Albee (1968) were

used to convert the raw intensity data to weight percent oxides. Analytical uncertainty is $1 - 2$ % for Si, Al, Fe, and K, $5 - 15$
% for Na, Mg, Ca, and Ti, and $50 - 100$ % for Mn, P, and Cr. EPMA analyses were performed as profiles across planar
deformation bands, at 3 to 14 μm intervals between each measurement depending on the thickness of the structure. To focus
on the analysis on the clay-size fraction of the sediment, the location of analysed spots were then visually checked to exclude
analyses in individual mineral grains. The remaining analyses were filtered to further exclude quartz or plagioclase ($SiO_2$

$+Al_2O_3 > 80$ wt% and sum of volatile elements $< 2$ wt%), calcite ($CaO > 40$ wt%), sulfides or oxides ($FeO$ or $TiO_2 > 30$ wt%)
and organic matter (sum of major elements $< 30$ wt%). This filtering resulted in the removal of 10 to 21 % of the data depending
on the profile.

After major element analyses, one of the samples (21R-2, 82-85) was further selected for in situ measurements of trace element
concentrations. Those measurements were conducted on the rock slab used for thin section preparation. In situ trace element

concentrations were determined at Montpellier 2 University on a ThermoFinnigan Element 2 High Resolution-Inductively
Coupled Plasma-Mass Spectrometer (HR-ICPMS) using a single collector double-focusing sector field Element XR (eXtended
Range) coupled with laser ablation (LA) system, a Geolas (Microlas) automated platform housing a 193 nm Compex 102 laser
from LambdaPhysik. The spot size was set to 102 μm, and therefore included multiple mineral grains. Oxide level, measured
using the ThO/Th ratio, was below 0.7%. Silicium 29 was used as internal standard. For each zone (matrix, shear zones and

veins), $^{29}Si$ was calibrated from the mean value of microprobe analyses in the same zone. Concentrations were calibrated
against the NIST 612 rhyolitic glass using the values given in Pearce et al. (1997). Data were subsequently reduced using the
GLITTER software using the linear fit to ratio method (Van Achterberg et al., 2001). This typically resulted in a 1 to 15%
precision (1sigma) for most analyses, evaluated by repeated analyses of reference basalt BIR before and after sample analysis
(Table A1 in Appendix). Detection limits were between < 1 and 50 ppb for most trace elements, between 0.04 and 0.5 ppm for

Li, B, Cr, Ti, Zn, and As, and between 1 and 15 ppm for Ni, Ca and Si.

## 4 Results

### 4.1 Petrological observation

The petrological inspection of those samples revealed the presence of authigenic pyrite and barite in the sediment. Pyrite is by
far the dominant authigenic mineral in the samples. Pyrite displays two crystal morphologies, cubic microcrysts and blocky

macrocrysts. Pyrite microcrysts (≤ 0.5 μm in diameter) occur as isolated crystals or as "framboids", i.e. circular aggregates, 2
- 20 μm in diameter (Fig. 3a-b). Framboids in deformation bands sometimes develop tails filled with pyrite microcrysts and





resembling pressure shadows (Fig. 3b-d), which may therefore be assimilated to porphyroclastic (grown before the deformation) or porphyroblastic (grown during the deformation) microstructures (Passchier and Trouw, 1998). Blocky pyrite occurs as macrocrysts (> 50 μm in diameter) with euhedral (but never cubic) or anhedral shapes (Fig. 3d). This blocky
morphology is scarce compared to the microcrystic and framboidal morphology, and occurs indifferently in the matrix and in deformation bands without a particular spatial distribution. Some blocky macrocrysts cut or seal the alignment of phyllosilicates in deformation bands (Fig. 3d), or grow at the expense of framboidal aggregates of microcrysts, which indicates a crystallization stage posterior to the deformation and also posterior to the microcrystic/framboidal morphology. As a general rule, pyrite is more abundant in deformation bands (up to 1.4% in volume) than in their host matrix (<0.5 vol%), the largest
modal proportions being found in veins (Table 2; Fig. 4). This pyrite enrichment is in fact caused by a greater abundance of microcrysts and framboids in the deformation bands than in the sediment matrix (Figs 3e-g, 5).

Barite has been observed only in one sample (21R-2, 82-85), as patches of ~5 μm long flakes filling a vein, with pyrite microcrysts and framboids growing on their faces (Fig. 3h).

### 4.2 X-ray diffraction

Ethylene glycol solvated XRD spectra of the shear zone and its host matrix (sample 4R-3, 73-76) are presented in Fig. 6. The different peaks reveal the presence of chlorite, kaolinite, quartz, feldspar and possibly vermiculite in both the shear zone and the matrix. A broad peak at 17 Å (5°2θ) indicates the presence of smectite or illite/smectite (I/S) mixed layers in the matrix, whereas this peak is absent in the shear zone. Illite is also present in both spectra as evidenced by the sharp peak at 10 Å (9°2θ). However, the width at half height of the 10 Å peak is smaller in the shear zone (0.14°2θ) than in the matrix (0.12°2θ), indicating
a greater crystallinity of illite in the shear zone than in its host sediment (Kübler, 1968).

### 4.3 Element repartitions

Two examples of XRF element maps, one in a shear zone and one in a vein, are provided in Fig. 7. An example of SEM element map, displaying a shear zone cutting a vein, is also provided in Fig. 5. In addition, two examples of EPMA profiles, one in a shear zone and one in a vein, are presented in Fig. 8. Averaged values of major element analyses are reported in Table
3 and presented in Fig. 9.

A common feature of all the maps and analyses is that major element concentrations are generally higher in the deformation bands than in the host matrix, indicating a higher density of the material in them. This is in particular true for Al, Si, Fe, K and S (Figs 7, 9). Within a given sample, the sum of elements is 11 – 16% higher in shear zones and 10 – 28% higher in veins than in the matrix, the highest values being reached in veins (Fig. 8, Table 3). The averaged sum of major elements is in the range
54.95 – 63.49 wt% for the clay-size fraction in the sediment matrix, whereas it is in the range 65.07 – 71.45 wt% for the clay-size fraction in shear zones and 66.68 – 72.91 wt% for the clay-size fraction in veins (Table 3). The comparison of XRF maps (Fig. 7) and EPMA analyses (Fig. 9) shows that the enrichment of Al, Si and K observed in deformation bands is matched by





an enrichment of these elements in the clay-size fraction of deformation bands. There are, however, noticeable exceptions to this general increase in major element concentrations: the Fe enrichment of deformation bands observed in element maps (Figs 5, 7) is not observed in the clay-size fraction (Fig. 9). This, and the correlation between Fe and S SEM maps (Fig. 5), shows that the Fe enrichment is due to the preferential growth of authigenic pyrite in deformation bands and does not occur in the clay-size fraction. Calcium is heterogeneously distributed in shear zones and is depleted in veins relative to the matrix (Figs 7, 9). Sodium, Ti, and Mg do not show any obvious variation between the matrix and deformation bands (Fig. 9).

Trace element analyses are reported in Table 4 and represented as averaged values in Fig. 10. As for the major elements, a general increase in trace element concentrations is observed in shear zones and in veins relative to the sediment matrix. For some elements, this enrichment is due to the use of Si as an internal standard, hence with an increased concentration in deformation bands as measured by EPMA (Table 3). Again, however, there are significant anomalies, among which a strong As enrichment in shear zones (+ 90%) and to a lesser extent in veins (+ 40%), a depletion of Ba in all the deformation bands (-31 to -33%), and an enrichment of Li and B in veins (+ 25 to + 38%).

## 5 Discussion

### 5.1 Compaction

The first result of our study is that deformation bands are more compacted than the host sediment. This greater compaction is seen in the SEM and XRF maps. It is also apparent in the EPMA data, as the concentration of volatile elements (approximated by 100 minus the sum of elements) is of 36.51 – 45.05 wt% in the matrix whereas it is only of 28.55 – 34.93 wt% in shear zones and 27.09 – 33.32 wt% in veins (Table 3), suggesting smaller interfoliar spaces in clays from deformation bands than from the matrix, and thus a smaller porosity. Shear zones and veins have therefore resulted in a greater volatile loss than the matrix, which may be interpreted as a greater fluid expulsion (Fig. 8). This result confirms the conclusion of *Milliken and Reed* (2010) that deformation microstructures in mud core samples from Site C0008 (IODP Exp. 316) formed primarily by mechanical compaction of the unconsolidated mud, and thus are indeed "dewatering structures". This strain-induced reduction of pore spaces and density increase is likely responsible for the observed enrichment of many major and trace elements in deformation bands compared to those of the matrix. However, mechanical compaction should raise all the analyzed elements by the same proportion. This hypothesis may be tested using a correction for compaction by normalizing the sum of element concentrations to 100 %. Even after such correction, some elements like As and to a lesser extent Cu, Sb and Pb remain enriched in deformation bands relative to the matrix and positively correlated to each other, while some like Ba are depleted in deformation bands (Figs 10, 11). Veins also appear enriched in B, Li and perhaps $K_2O$ but slightly depleted in CaO and perhaps $Na_2O$ relative to the matrix. This indicates that some chemical reactions occurred in the deformation bands, in addition to mechanical compaction. In the following, we explore the processes that may be responsible for the chemical differences observed in shear zones and veins relative to their host sediment matrix.



## 5.2 Deformation and pyrite diagenesis

The dark color of deformation bands (Fig. 2), their increased content in authigenic pyrite microcrysts (Figs 3e-g, 4, 5), their enrichment in chalcophile elements (Cu; As; Sn; Pb, Figs 10, 11) and the positive correlations among these elements (Fig. 11), all concur to indicate that sulfide precipitation was enhanced in deformation bands compared to the matrix. Sulfide mineralization in low-temperature ($< 60°C$) sediments is essentially a byproduct of the anaerobic degradation of organic matter by microorganisms (see review by Megonigal et al. 2004). For this reason, authigenic pyrite is generally observed in anoxic,

organic matter-rich sediments such as shales. The framboidal morphology of pyrite itself is generally, though not exclusively, taken as an indicator of microbial proliferation (Barbieri and Cavalazzi, 2005; Cavalazzi et al., 2012; Chen et al., 2006; Merinero et al., 2009; Peckmann et al., 2001). Organic carbon decomposition by heterotrophic microorganisms has the effect of releasing reduced dissolved iron and hydrogen sulfide, through $Fe^{3+}$ and sulfate reduction reactions such as:

$$CH_2O + 4Fe(OH)_3 + 7CO_2 \rightarrow 8HCO_3^- + 4Fe^{2+} + 3H_2O \qquad (1)$$

$$2CH_2O + SO_4^{2-} \rightarrow 2HCO_3^- + H_2S \qquad (2)$$

where $CH_2O$ represents organic matter. $CO_2$ in equation (1) may be supplied by methanogenesis:

$$2CH_2O \rightarrow CO_2 + CH_4 \qquad (3)$$

The products of these reactions are then involved in a suite of secondary redox reactions, among which pyrite ($FeS_2$) precipitates via the formation of temporary sulfide phases (Hunger and Benning, 2007; Wilkin and Barnes, 1997):

$$Fe^{2+} + 2HS^- \rightarrow FeS_2 + H_2 \qquad (4)$$

The precipitation of multiple pyrite microcrysts forming framboids requires a nucleation rate significantly greater than the crystal growth rate (i.e. surface-controlled growth), a condition achieved as long as the supply of reactants is not limited and the thermodynamic conditions are far from equilibrium (Ohfuji and Rickard, 2005; Wilkin and Barnes, 1997). This suggests that sulfate and reduced iron supplies were not limited at the time of deformation, and thus that the deformation bands occurred

within the sulfate reduction zone. The Ba depletion in deformation bands (Figs 10, 11) and the growth of pyrite microcrysts and framboids on barite needles in a vein (Fig. 3h) also support sulfate bio-reduction via metabolic reactions (1) and (2). All these lines of evidence lead to the conclusion that the preferential crystallization of microcrystic and framboidal pyrite in deformation bands is a result of enhanced bacterial activity. According to syn-deformation microstructures of pyrite growth (Fig. 3b, c), this enhanced bacterial activity was coeval with the development of deformation bands and the associated reduction

of porosity.

On the contrary, the blocky morphology of large and rare pyrite crystals indicates a transport-controlled crystal growth (Ohfuji and Rickard, 2005; Wilkin and Barnes, 1997), meaning that the demand of sulfate and iron exceeded their supply. These large blocky pyrite crystals grow at the expense of framboids and seal the CPO of deformation bands (Fig. 3d). This suggests that blocky pyrite is a recrystallization form of the framboid morphology, occurring without deformation, below the

sulfate/methane transition zone, and hence not necessarily via metabolic reactions.





## 5.3 Clay transformation

In tandem with pyrite diagenesis, XRD spectra and chemical analyses show that the development of deformation bands is accompanied by modifications of the clay mineralogy. In the only sample studied by XRD, the shear zone displays a disappearance of smectite or I/S mixed layers, and an increased crystallinity of illite, relative to its host matrix. In the samples

analyzed for trace elements, the correlated B and Li enrichments relative to the matrix, both particularly noticeable in veins (Fig. 11), are two additional arguments strongly suggesting that deformation bands localize smectite transformation into illite. Indeed, illitization is known to result in an uptake of B and Li, as the former element substitutes in tetrahedral sites of illite and the later in octahedral sites (Williams et al., 2013).

Because of its importance as a reaction releasing freshwater, the destabilization of smectite and its transformation into illite

has received a large attention for the fluid budget of accretionary prisms. Heat and the availability of $K^+$ are considered as the primary factors governing illitization, and time, pressure or shear stress as secondary factors (Casciello et al., 2011; Ransom and Helgeson, 1995). Given the small size of the microstructures studied here, however, it is quite unlikely that any of these factors varied much between deformation bands and their matrix just a few millimeters apart. Biotic alteration may offer an alternative, more plausible explanation to the observed changes in the mineralogy of clays. Indeed, smectite and I/S mixed

layers are mineral structures very sensitive to biotic alteration, because they are targets for anaerobic bacteria in order to reach structural $Fe^{3+}$ (Dong et al., 2009; Esnault et al., 2013; Zhang et al., 2012). There are strong suspicions that microbial reduction of smectite and I/S mixed layers is able to trigger illitization (Esnault et al., 2013; Koo et al., 2014, 2016). Accordingly, we propose that enhanced anaerobic bacterial proliferation in deformation bands, a process already suggested by the localized crystallization of framboidal pyrite, is also at the origin of the disappearance of smectite or I/S mixed layers, and the increased

crystallinity of illite.

## 5.4 Causes of deformation-enhanced bacterial proliferation

A question that arises from the above discussion is why did the deforming sediment provide a more favorable ground to the development of anaerobic microorganisms than the undeformed matrix? A first possible explanation is that the greater compaction of deformation bands increased the availability of compounds necessary for metabolic reactions (1) and (2) by

increasing their concentration compared to the matrix. As shown above, the majority of element concentrations are raised by ~20% due to the increased compaction of deformation bands (Figs 8, 10), and thus the same enrichment is expected for organic matter and $Fe^{3+}$. This concentration increase may have favored the proliferation of anaerobic microorganisms.

Alternatively or additionally, it has been shown that the deformation of silicate minerals, and the delamination of clay layers in particular, generate $H_2$ by chemical reactions between water and mechanoradicals created by the rupture of Si–O–Si or Al–

O–Si bonds (Hirose et al., 2011; Kameda et al., 2004; Kita et al., 1982; Wakita et al., 1980). This $H_2$ may have served as a terminal electron donor in heterotrophic metabolic reactions, acting in conjunction with mechanical compaction to favor anaerobic microbial proliferation during deformation of the unconsolidated sediment.





### 5.5 Implications for the deformation of mudstones in accretionary prisms

Reports of all the drilling expeditions across the Nankai accretionary prism are consistent in describing the ubiquitous existence

of dark shear zones and veins in mud sediments (Ashi et al., 2008; Henry et al., 2012; Kinoshita et al., 2009a, 2009b, 2009c, 2009d, 2009e, 2009f; Saffer et al., 2010; Ujiie et al., 2004). Dark deformation bands have also been described onland in mudstones of the Boso peninsula paleo-accretionary prism (Ohsumi and Ogawa, 2008), and worldwide in active continental margins (e.g. Behrmann et al. 1988), sometimes in conjunction with an enrichment in authigenic iron sulfides (Lindsley-Griffin et al., 1990). Dark deformation bands may thus be considered as an intrinsic feature of mudstone sediments in accretionary

prisms. Our results show that these small structures localize pyrite crystallization and smectite-illite transformations, two diagenetic reactions probably mediated by the proliferation of anaerobic microorganisms, and both boosted by deformation. A possible implication of our study is that such increased diagenesis in deformation bands may be a source of fresh water, firstly because $H_2O$ is a product of metabolic reactions leading to pyritization (see Eqn. (1) for instance), and secondly because $H_2O$ is also released by the illitization of smectite. In pervasively deformed mudstones with abundant deformation bands, these two

effects may combine to potentially explain the local deficits of modeled versus observed pore water freshening found in previous studies (e.g. Saffer and McKiernan 2009). More work is obviously necessary to quantify the contribution of microbial diagenesis to the pore water freshening of deformed sediments.

Another important implication of our study is that the microbial diagenesis might be a general feature of deformation structures whatever their size, from the tiny deformation bands described here to major thrust faults. This is particularly the case of the

megasplay fault of the Nankai prism drilled at 270 mbsf at site C0004 during the NanTroSEIZE expedition 316 (Kinoshita et al., 2009c). The analysis of the core zone of this megasplay revealed a 2 cm-thick dark gouge, interpreted as the principal slip zone and the subject of a vigorous scientific controversy. Indeed, this dark gouge was found to combine an increased vitrinite reflectance, a smectite depletion, and an increased illite crystallinity compared to the surrounding breccia (Sakaguchi et al., 2011; Yamaguchi et al., 2011). These differences were interpreted as thermal maturation of organic matter and illitization in

the dark gouge, both due to coseismic shear heating above 380°C in the principal slip zone of the megasplay, and thus as an evidence of seismic rupture propagation to the sea floor. However, other studies using trace element analyses as well as thermal modeling contradicted this interpretation, by concluding that the coseismic temperature rise in the megasplay did not exceed 300°C, and could not be sufficient to activate the kinetics of illitization (Hirono et al., 2009, 2014). How can these two divergent views of the fault zone chemistry and mineralogy be reconciled?

The paradox might be solved by considering bacterial activity as a diagenetic process boosted by deformation. It is important to note that a peak bacterial concentration of $3.6 * 10^9$ cells/cm$^3$ was found in the damaged zone of the megasplay (Kinoshita et al., 2009c). This cell abundance, noticeably in the upper range of sediments worldwide (typically of $10^6 - 10^9$ cells/cm$^3$), was also nearly as high as that of the sea floor at this site C0004. Moreover, the lower part of the dark gouge in the megasplay fault drilled at site C0004 was also enriched in authigenic pyrite (Destrigneville et al., 2013), which makes diagenetic reactions

in the megasplay very similar to those found in the tiny deformation bands studied here. Such an analogy strongly suggests





that temperature rise is not the only factor capable of explaining the observed anomalies in the megasplay, and that many mineral reactions in the principal slip zone might be biologically mediated rather than thermally activated. Recent studies have shown that anaerobic microbial activity associated with framboidal pyrite crystallization could locally yield anomalously high reflectance values of amorphous organic matter compared to thermal maturation (e.g. Synnott et al. 2016). Microbial activity

could thus explain the elevated vitrinite reflectance temperature obtained for the principal slip zone (Sakaguchi et al., 2011), as it does explain pyrite crystallization, smectite or I/S mixed layers destabilization, and illitization in deformation structures whatever their size. Anaerobic bacterial bloom, boosted by deformation, would thus solve the paradox of mineralogical and chemical anomalies in the principal slip zone of the Nankai megasplay. We therefore come to the conclusion that more caution is required before interpreting illitization in fault gouges as an evidence of coseismic slip, or more generally as a temperature

rise, because this mineral reaction, as others, might be triggered or enhanced by microbial activity.

## 6 Conclusions

Microscopic observation, X-ray diffraction, and major-trace element analyses reveal that deformation bands in mud sediments of the Nankai accretionary prism, which are similar to microstructures found worldwide in active and passive margin sediments, have localized some diagenetic reactions in addition to mechanical compaction. The first diagenetic reaction is an

increased precipitation of microcrystic and framboidal pyrite. The second diagenetic reaction found in deformation bands is a change in the mineralogy of clays compared with the surrounding sediment, consisting in the destabilization of smectite or I/S mixed layers and the crystallization of illite. Both diagenetic reactions may be explained by a locally enhanced activity of anaerobic microorganism in deformation bands, which may be related to the generation of $H_2$ by intracrystalline deformation of silicates minerals. This dual biogenic diagenesis occurred essentially during deformation and vanished afterwards.

Biologically-induced diagenetic reactions detected in deformations bands may have consequences on the fluid budget of deforming sediments, because biogenic pyrite crystallization and illitization produce freshwater, which may participate in reducing the chlorinity of pore waters from accretionary prisms. These finding might also affect our understanding of coseismic reactions in fault zones, because illitization, usually taken as an indicator of temperature rise, might be mediated by metabolic processes rather than by shear heating.



**Appendix A**

|        | BIR    | BIR    | BIR    | BIR    | average | 1 s.d. |
|--------|--------|--------|--------|--------|---------|--------|
| Li7    | 2,94   | 2,87   | 2,96   | 2,82   | 2,90    | 0,06   |
| B11    | 1,06   | 1,20   | 1,13   | 1,01   | 1,10    | 0,08   |
| Si*    | 20,10  | 20,10  | 20,10  | 20,10  | 20,10   | 0,00   |
| Ca*    | 7,75   | 7,79   | 8,15   | 8,05   | 7,93    | 0,20   |
| Sc45   | 33,72  | 33,87  | 37,06  | 36,75  | 35,35   | 1,80   |
| Ti*    | 0,55   | 0,55   | 0,55   | 0,55   | 0,55    | 0,00   |
| V51    | 285,61 | 284,11 | 277,50 | 280,45 | 281,92  | 3,66   |
| Cr53   | 376,74 | 385,38 | 360,64 | 365,20 | 371,99  | 11,21  |
| Co59   | 48,21  | 48,04  | 47,64  | 47,77  | 47,92   | 0,26   |
| Ni62   | 160,96 | 159,61 | 158,43 | 157,04 | 159,01  | 1,67   |
| Cu63   | 109,51 | 107,63 | 108,30 | 109,09 | 108,63  | 0,84   |
| Zn66   | 71,19  | 67,62  | 68,17  | 69,14  | 69,03   | 1,57   |
| Zn68   | 58,87  | 56,79  | 59,15  | 60,01  | 58,71   | 1,37   |
| As75   | 0,11   | 0,10   | 0,06   | 0,06   | 0,08    | 0,03   |
| Rb85   | 0,22   | 0,19   | 0,20   | 0,21   | 0,20    | 0,01   |
| Sr88   | 85,46  | 85,68  | 88,90  | 88,24  | 87,07   | 1,76   |
| Y89    | 10,34  | 10,28  | 11,72  | 11,33  | 10,92   | 0,72   |
| Zr90   | 9,53   | 9,49   | 10,73  | 10,28  | 10,01   | 0,60   |
| Nb93   | 0,41   | 0,41   | 0,43   | 0,42   | 0,41    | 0,01   |
| Sb121  | 0,57   | 0,57   | 0,55   | 0,55   | 0,56    | 0,01   |
| Cs133  | b.d.l. | 0,00   | b.d.l. | b.d.l. | 0,00    | 0.002  |
| Ba137  | 5,56   | 5,34   | 5,45   | 5,50   | 5,46    | 0,09   |
| La139  | 0,48   | 0,49   | 0,50   | 0,50   | 0,49    | 0,01   |
| Ce140  | 1,67   | 1,67   | 1,68   | 1,73   | 1,69    | 0,03   |
| Pr141  | 0,30   | 0,31   | 0,32   | 0,32   | 0,31    | 0,01   |
| Nd146  | 1,90   | 1,93   | 2,00   | 2,00   | 1,96    | 0,05   |
| Sm147  | 0,85   | 0,84   | 0,90   | 0,90   | 0,87    | 0,03   |
| Eu151  | 0,40   | 0,41   | 0,43   | 0,43   | 0,41    | 0,01   |
| Gd157  | 1,33   | 1,28   | 1,52   | 1,44   | 1,39    | 0,11   |
| Tb159  | 0,25   | 0,25   | 0,29   | 0,26   | 0,26    | 0,02   |
| Dy163  | 1,91   | 1,89   | 2,11   | 2,13   | 2,01    | 0,13   |
| Ho165  | 0,43   | 0,43   | 0,46   | 0,47   | 0,45    | 0,02   |
| Er167  | 1,22   | 1,23   | 1,39   | 1,34   | 1,30    | 0,08   |
| Tm169  | 0,19   | 0,19   | 0,21   | 0,20   | 0,20    | 0,01   |
| Yb173  | 1,33   | 1,27   | 1,44   | 1,42   | 1,36    | 0,08   |
| Lu175  | 0,19   | 0,18   | 0,21   | 0,20   | 0,20    | 0,01   |
| Hf177  | 0,40   | 0,40   | 0,45   | 0,45   | 0,42    | 0,03   |
| Ta181  | 0,03   | 0,03   | 0,04   | 0,03   | 0,03    | 0,00   |
| Pb208  | 3,48   | 3,49   | 3,62   | 3,45   | 3,51    | 0,08   |
| Th232  | 0,02   | 0,02   | 0,03   | 0,03   | 0,03    | 0,00   |
| U238   | 0,02   | 0,02   | 0,01   | 0,02   | 0,02    | 0,00   |

**Table A1: Trace element composition of reference basalt BIR repeatedly measured by HR-LA-ICPMS at the beginning and the end of the analytical session (all elements are in ppm, except Si*, Ca* and Ti* in wt%).**

**Author contribution**

VF chose the samples during IODP Exp. 315. AMB and VF did the petrological inspection of samples. HR, MA and VF carried out the mineralogical and chemical analyses. VF prepared the manuscript with contributions from all co-authors.



**Competing interests**

The authors declare that they have no conflict of interest.

**Acknowledgements**

We thank the D/V Chikyu staff for their support during IODP Expedition 315. We also thank our colleagues Timothy. B. Byrne, Jonathan C. Lewis, Kyuichi Kanagawa, and Jan Behrmann for our onboard discussions about the nature of deformation bands. This research used samples and data provided by the Integrated Ocean Drilling Program. Funding was provided by INSU-CNRS "Sismofluids" and "3F" grants.

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



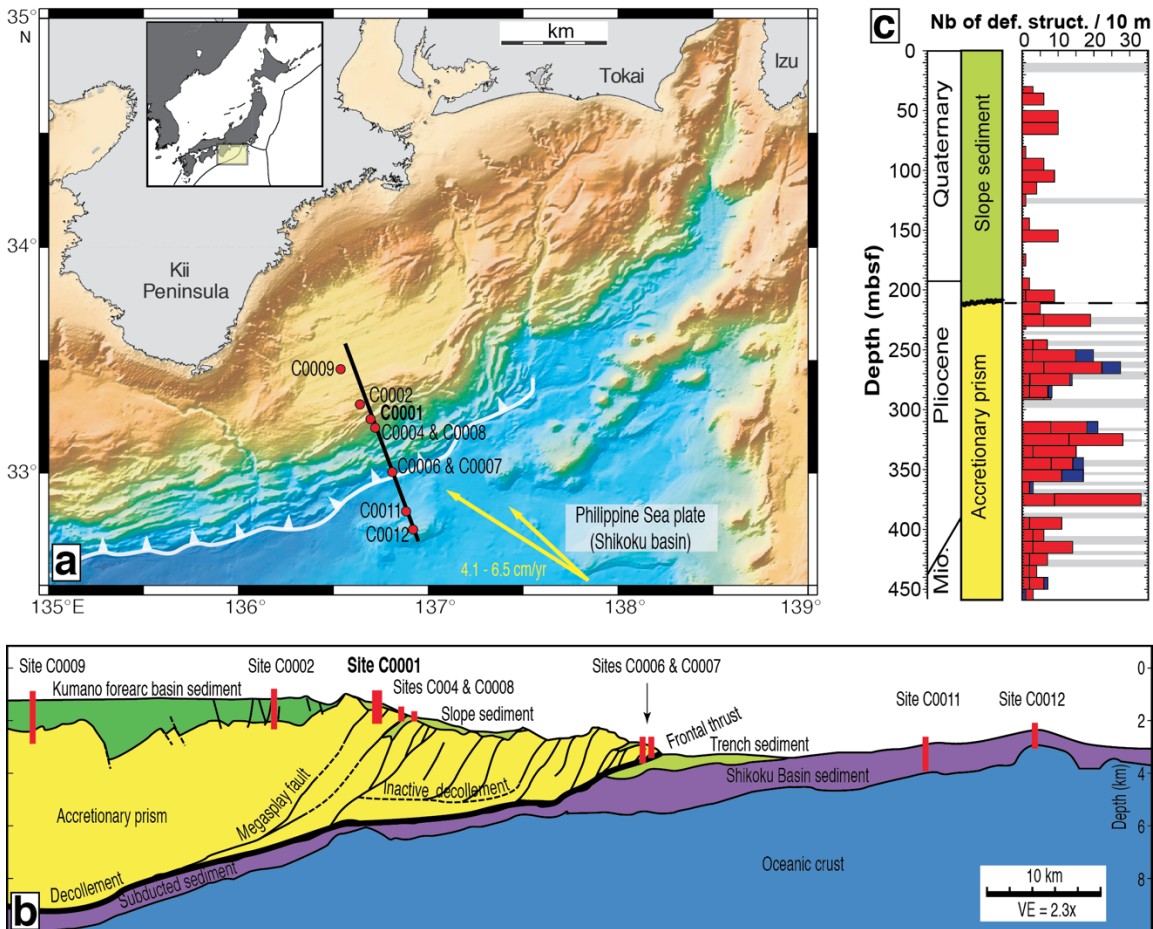

**Figure 1: a)** Bathymetric map of the Nankai accretionary prism offshore of the Kii peninsula, showing the location of the core logging sites of the NanTroSEIZE transect (IODP Exp. 315, 316, 319, 322, and 333). The yellow arrows represent the range of plate convergence directions (modified from Moore et al., 2009). **b)** Schematic cross section of the Nankai accretionary prism along the NanTroSEIZE transect (modified from Moore et al., 2009). **c)** Simplified stratigraphy and histogram of deformation bands (10 m spacing) found in cores from site C0001 (modified from Kinoshita et al., 2009). The slope sediments (Unit I) and sediments from the accretionary prism (Unit II) are separated by an unconformity labelled by a thick black line. Red bars stand for shear fractures (shear zones + faults), blue bars for veins, and grey bars correspond to zones of no core recovery.





**Figure 2: Subvertical deformation bands observed in cores of the NanTroSEIZE transect, showing a darker color than the matrix and a crystal preferred orientation. S0 represents the stratification plane. a) and b) Macroscopic and microscopic (cross-polarized light) views of a shear zone with a normal component of slip (sample C0001 4R-3, 73-76). c) and d) Macroscopic and microscopic (cross-polarized light with a lambda plate) views of anastomosing veins (sample C0001 10R-2, 35-40). Core top is upward on all the pictures. Scale bar is 1 cm in macroscopic pictures.**







**Figure 3: SEM pictures of pyrite textures. a) Pyrite framboid made of aggregated cubic pyrite microcrysts in a shear zone (sample 21R-2, 82-85). b) Pyrite framboid in a shear zone, with pressure shadow-like tails of isolated microcrysts (4R-3, 73-76), showing that the microcryst morphology grew before and/or during deformation. c) Same as b) with a lower magnification. d) Blocky euhedral**
**pyrite macrocryst embedding particles oriented parallel to the crystallographic preferred orientation (CPO) in a shear zone, showing that blocky pyrite growth is posterior to the CPO (4R-3, 73-76). e), f) and g) Pyrite framboids and isolated cubic pyrite microcrysts in a shear zone (21R-4, 89-93). Arrows delimit the boundary between the shear zone (with a CPO of particles) and the matrix (with randomly-oriented particles). Note the greater abundance of pyrite crystals in the shear zone compared to the matrix. h) Barite flakes surrounded by pyrite microcrysts and framboids in a vein. A wider view of the structure is provided in Figure 5 (21R-2, 82-**
**85).**

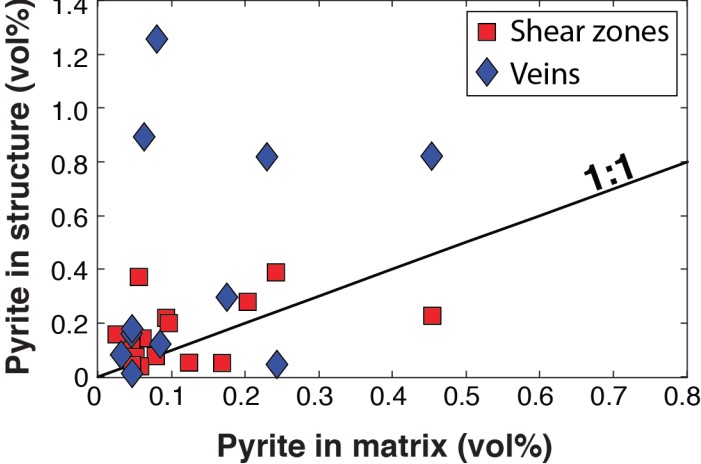

**Figure 4: Modal proportions of pyrite in deformation bands (shear zones and veins) relative to their host matrix (data provided in Table 2).**

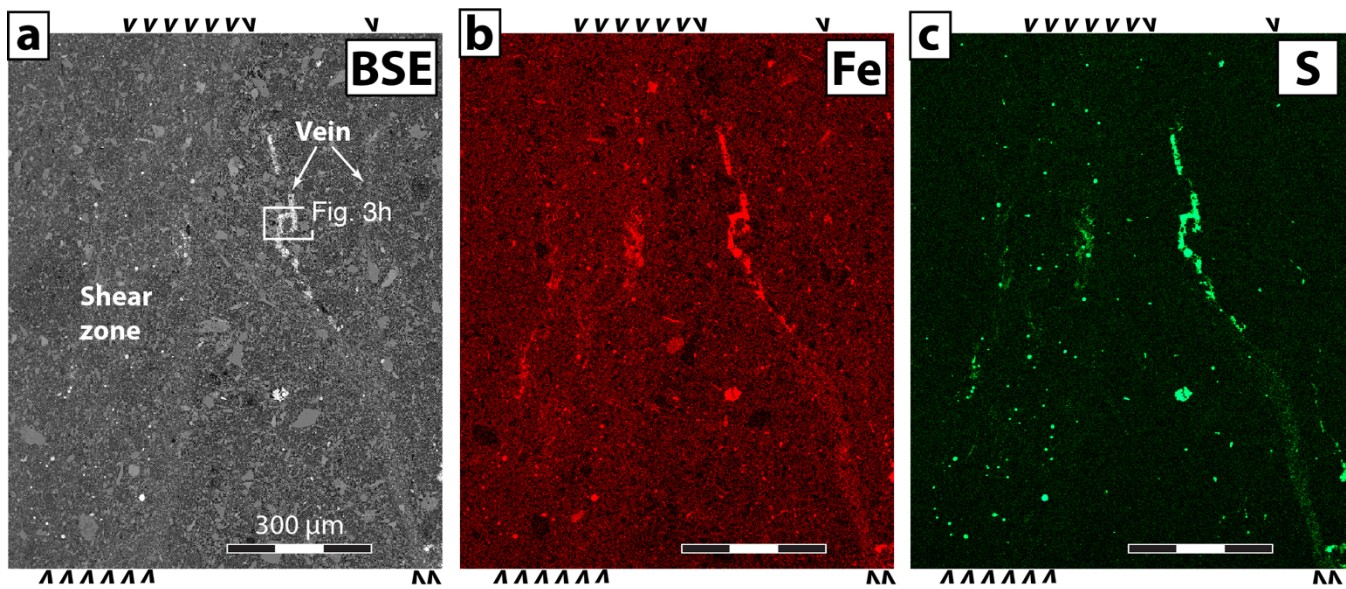

**Figure 5: Example of SEM map in a shear zone and a vein (sample 21R-2, 82-85), showing the greater modal abundance of pyrite crystals in deformation bands (represented by arrows) relative to the host sediment matrix. a) back-scattered electron picture. b) and c) Fe and S maps, respectively.**





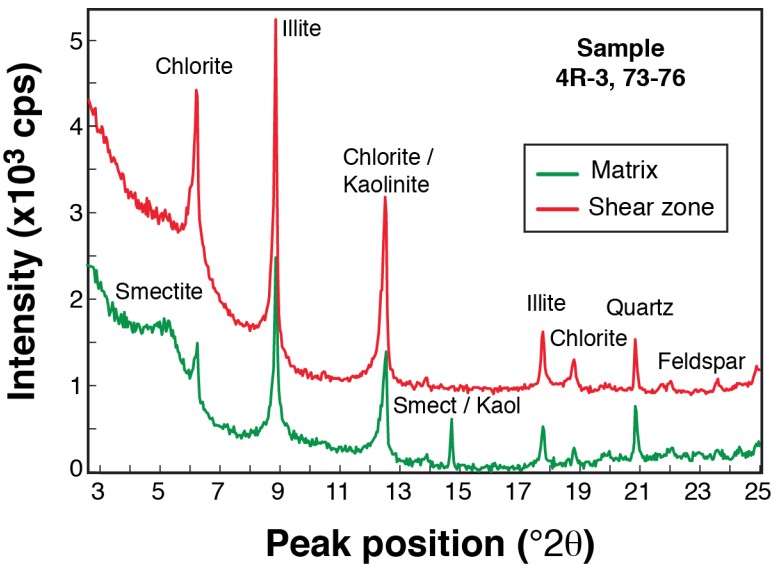

**Figure 6: Powder X-ray diffraction spectra of the ethylene-glycol saturated clay-size fraction extracted from a shear zone (in red) and its host matrix (in green) in sample 4R-3, 73-76.**









**Figure 7: Examples of XRF maps in a shear zone (sample 21R-2, 82-85) and a vein (10R-2, 2-12). a) Optical microphotographs (plain light). b) to f) Al, Si, Ca, Fe, and K maps, respectively. Note the enrichment in all the analyzed elements relative to the matrix, except for Ca that is heterogeneously distributed in the shear zone and in the matrix (due to the drag of Ca-rich and Ca-poor beddings) and not enriched in the vein.**

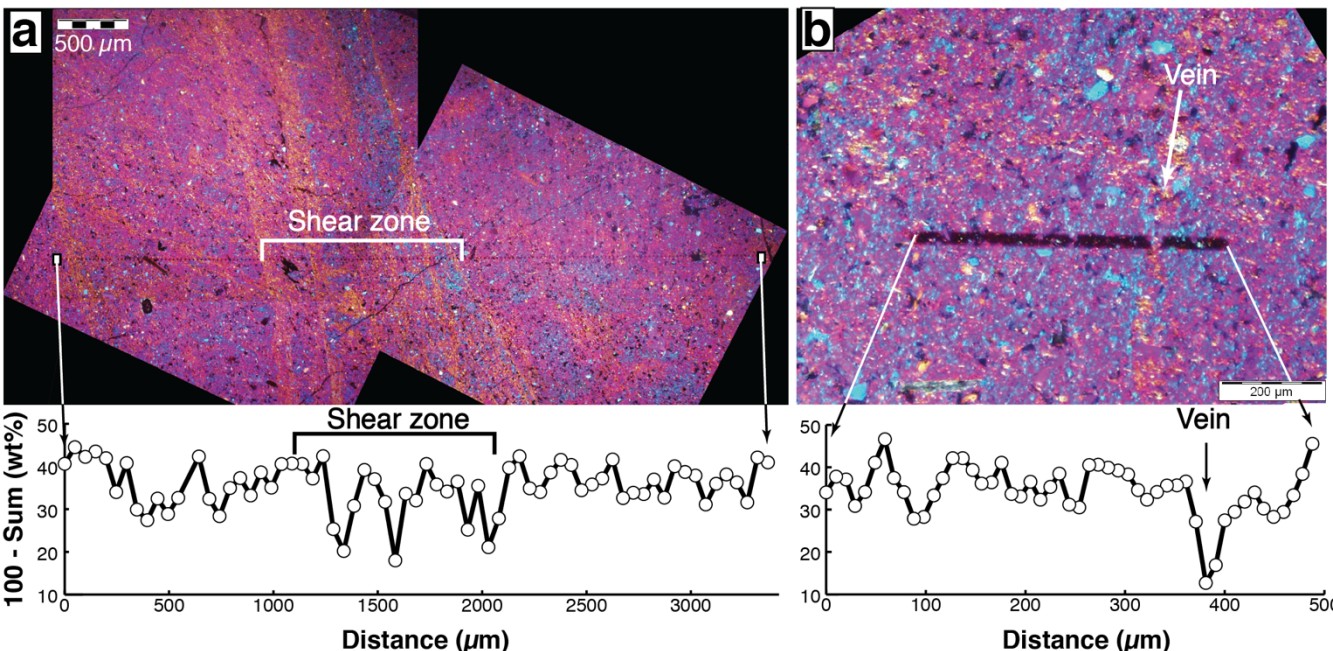

**Figure 8: Examples of EPMA profiles of major element analyses in the clay-size fraction of deformation bands, reported onto optical microphotographs (cross-polarized light with a lambda plate). The quantity displayed in the profiles (100 minus the sum of analyzed major elements) is assumed to represent the concentration of volatile (i.e. not analyzed) elements. a) Shear zone (sample 4R-3, 73-76). b) Vein (21R-2, 82-85).**



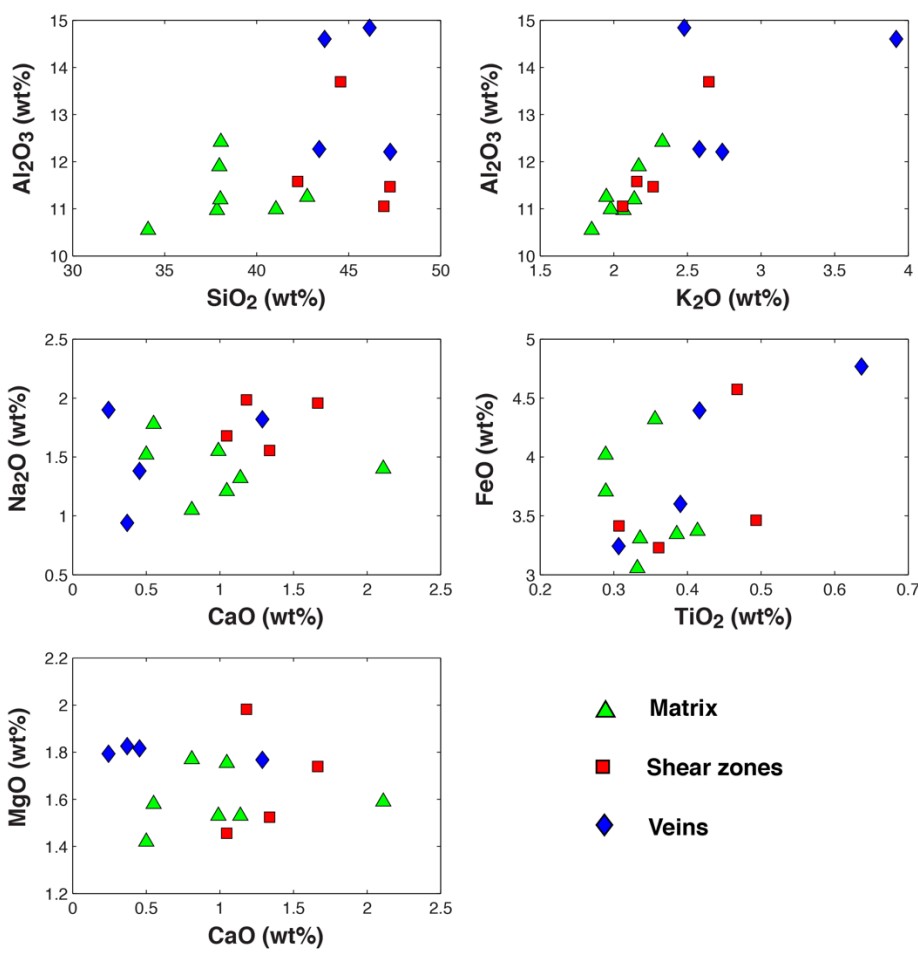

**Figure 9: Averaged major element concentrations of EPMA analyses in the clay-size fraction of shear zones, of veins and of their host sediment matrix. Data provided in Table 3.**





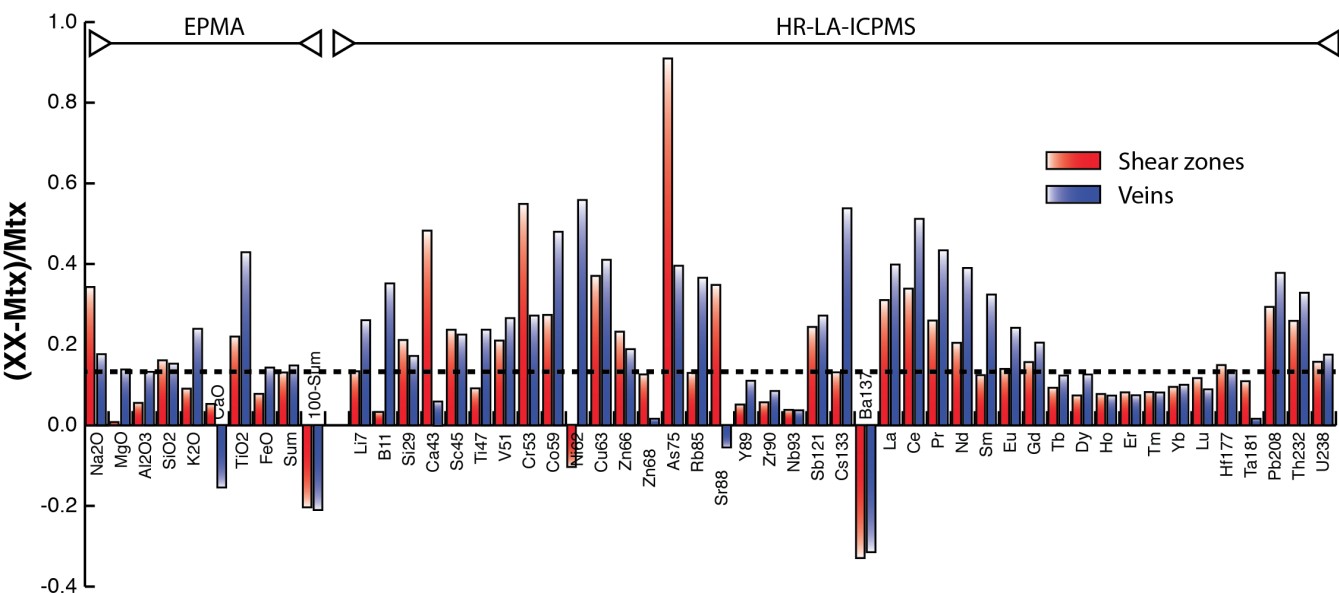

**Figure 10: Average relative proportions of major element concentrations (measured by EPMA on samples 4R-3, 73-76; 11R-6, 3-8; and 21R-2, 82-85) and trace element concentrations (measured by HC-LA-ICPMS on sample 21R-2, 82-85) in shear zones (red bars) and in veins (blue bars) with respect to their host sediment matrix. The dashed horizontal line corresponds to the averaged enrichment of major elements from the matrix (Mtx) to the deformation bands (XX), assumed to represent mechanical compaction.**

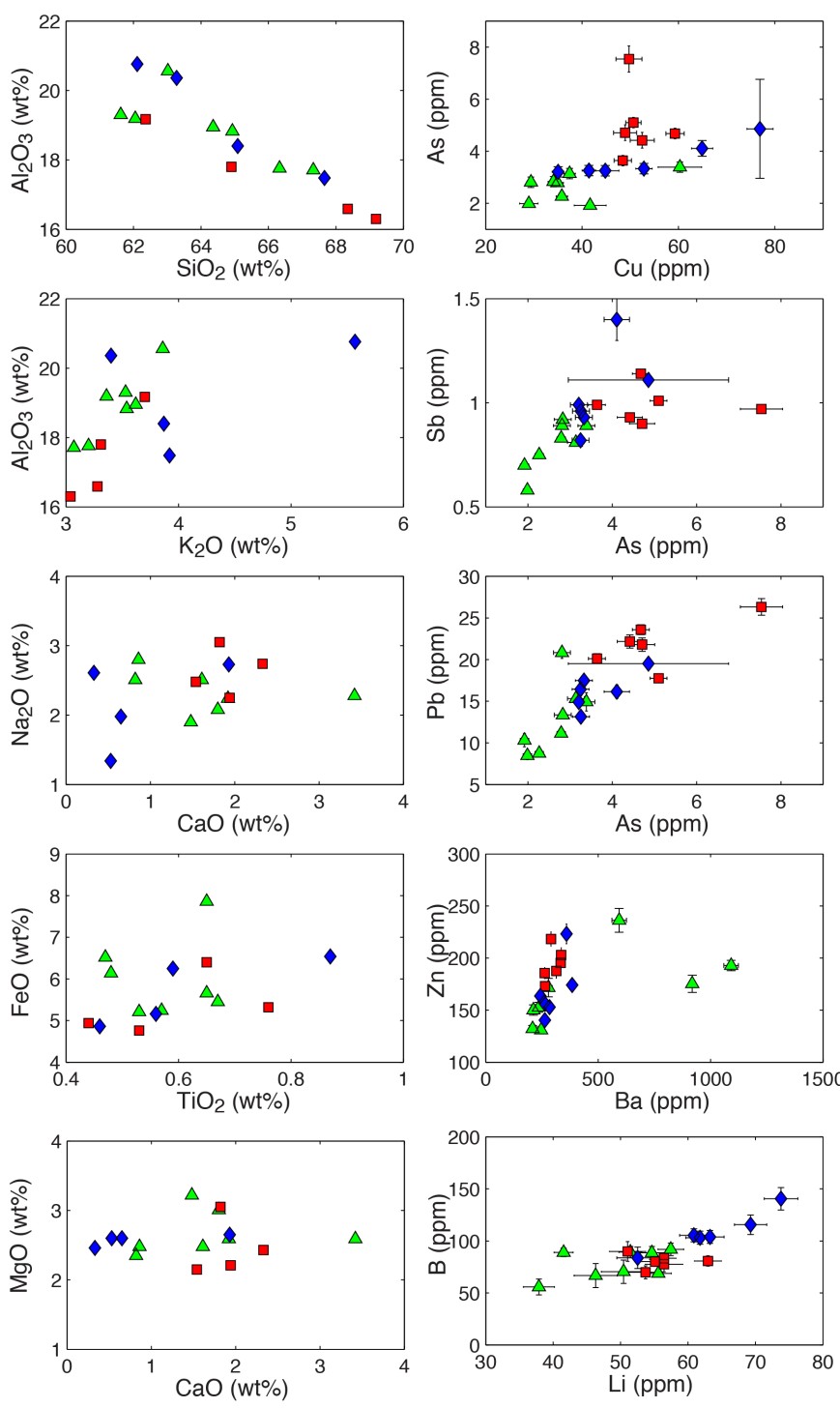

**Figure 11: Averaged major element concentrations represented as oxides (samples 4R-3, 73-76; 11R-6, 3-8, and 21R-2, 82-85) and trace element concentrations (21R-2, 82-85) in the matrix (green triangles), in shear zones (red squares), and in veins (blue diamonds) after normalization of the sum of major oxides to 100%.**






| Core-section, interval (cm) | Depth (mbsf) | Thin section | Observed structures | SEM | XRD | XRF | EPMA | LA-ICPMS |
|---|---|---|---|---|---|---|---|---|
| 2R-5, 59-64 | 245.2 | VFC19 | Shear zone | | | | | |
| 2R-5, 102-107 | 245.7 | VFC18 | None | | | | | |
| 4R-3, 48-50 | 261.8 | VFC14 | Shear zone + veins | | | | | |
| 4R-3, 73-76 | 262 | VFC6 | Shear zone + veins | X | X | | X | |
| | 262 | VFC7 | Shear zone | | | | | |
| 5R-4, 30-34 | 271.1 | VFC22 | Veins | | | | | |
| | 271.1 | VFC23 | Shear zone + veins | | | | | |
| 10R-2, 2-12 | 313.4 | VFC15 | Veins | | | X | X | |
| | 313.4 | VFC16 | Veins | | | | | |
| 10R-2, 35-40 | 313.7 | VFC8 | Veins | | | | | |
| | 313.7 | VFC9 | Shear zone | | | | | |
| 10R-3, 31-39 | 315.1 | VFC1 | Veins | | | | | |
| 10R-4, 37-44 | 316 | VFC25 | Shear zone | | | | | |
| 11R-2, 59-61 | 323.5 | VFC10 | Shear zone | | | | | |
| | 323.5 | VFC11 | Shear zone | | | | | |
| 11R-6, 3-8 | 327.4 | VFC17 | Shear zone | | | | X | |
| 12R-3, 93-97 | 334.7 | VFC2 | Shear zone | | | | | |
| 13R-1, 137-140 | 341.8 | VFC5 | Shear zone | | | | | |
| 13R-2, 80-88 | 342.7 | VFC20 | Veins | | | | | |
| | 342.7 | VFC21 | None | | | | | |
| 21R-2, 82-85 | 413.7 | VFC3 | Shear zone | X | | | X | X |
| | 413.7 | VFC4 | Shear zone + veins | X | | X | X | X |
| 21R-4, 89-93 | 416.2 | VFC12 | Shear zone | X | | | | |
| | 416.2 | VFC13 | Shear zone | | | | | |
| 21R-5, 50-53 | 417.3 | VFC24 | Veins | | | | | |

**Table 1: Summary of studied samples from cores at site C0001, of structures observed, and of analyses carried out on them.**





| Core-Section. interval (cm) | Depth (mbsf) | Thin section | Observed structures | Vol% Pyrite in matrix | Vol% Pyrite in shear zones | Vol% Pyrite in veins |
|---|---|---|---|---|---|---|
| 2R-5, 59-64 | 245.2 | VFC19 | Shear zone | 0.241 | 0.391 | - |
| 2R-5, 102-107 | 245.7 | VFC18 | None | 0.116 | - | - |
| 4R-3, 48-50 | 261.8 | VFC14 | Shear zone + veins | 0.085 | - | 0.125 |
| 4R-3, 73-76 | 262 | VFC6 | Shear zone + veins | 0.452 | 0.226 | 0.825 |
|  | 262 | VFC7 | Shear zone | 0.125 | 0.053 | - |
| 5R-4, 30-34 | 271.1 | VFC22 | Veins | 0.228 | - | 0.821 |
|  | 271.1 | VFC23 | Shear zone + veins | 0.08 | 0.08 | 1.26 |
| 10R-2, 2-12 | 313.4 | VFC15 | Veins | 0.032 | - | 0.083 |
|  | 313.4 | VFC16 | Veins | 0.044 | - | 0.161 |
| 10R-2, 35-40 | 313.7 | VFC8 | Veins | 0.174 | - | 0.294 |
|  | 313.7 | VFC9 | Shear zone | 0.056 | 0.037 | - |
| 10R-3, 31-39 | 315.1 | VFC1 | Veins | 0.061 | - | 0.892 |
| 10R-4, 37-44 | 316 | VFC25 | Shear zone | 0.055 | 0.374 | - |
| 11R-2, 59-61 | 323.5 | VFC10 | Shear zone | 0.059 | 0.14 | - |
|  | 323.5 | VFC11 | Shear zone | 0.169 | 0.052 | - |
| 11R-6, 3-8 | 327.4 | VFC17 | Shear zone | 0.052 | 0.074 | - |
| 12R-3, 93-97 | 334.7 | VFC2 | Shear zone | 0.044 | 0.046 | - |
| 13R-1, 137-140 | 341.8 | VFC5 | Shear zone | 0.025 | 0.159 | - |
| 13R-2, 80-88 | 342.7 | VFC20 | Veins | 0.243 | - | 0.044 |
|  | 342.7 | VFC21 | None | 0.292 | - | - |
| 21R-2, 82-85 | 413.7 | VFC3 | Shear zone | 0.203 | 0.28 | - |
|  | 413.7 | VFC4 | Shear zone + veins | 0.045 | 0.12 | 0.152 |
| 21R-4, 89-93 | 416.2 | VFC12 | Shear zone | 0.092 | 0.217 | - |
|  | 416.2 | VFC13 | Shear zone | 0.097 | 0.2 | - |
| 21R-5, 50-53 | 417.3 | VFC24 | Veins | 0.047 | - | 0.012 |

**Table 2: Modal proportions of pyrite in deformation bands (shear zones and veins) and in their host matrix deduced from the analysis of optical microscopic pictures.**

| Sample | 4R 3 73-76 | | | | | 11R 6 3-8 | | 21R 2 82-85 | | | | | | | |
|---|---|---|---|---|---|---|---|---|---|---|---|---|---|---|---|
| Structure | Matrix 1 | Shear zone 1 | Vein 1 | Matrix 2 | Shear zone 2 | Matrix | Shear zone | Matrix 1 | Shear zone 1 | Matrix 2 | Vein 2 | Matrix 3 | Vein 3 | Matrix 4 | Vein 4 |
| Nb of analyzes | 104 | 17 | 4 | 25 | 43 | 33 | 36 | 31 | 22 | 40 | 11 | 46 | 10 | 55 | 8 |
| Na2O | 1,21 | 1,98 | 1,82 | 1,32 | 1,68 | 1,40 | 1,96 | 1,55 | 1,56 | 1,78 | 1,38 | 1,52 | 1,90 | 1,05 | 0,94 |
| MgO | 1,75 | 1,98 | 1,77 | 1,53 | 1,46 | 1,59 | 1,74 | 1,53 | 1,52 | 1,58 | 1,82 | 1,42 | 1,79 | 1,77 | 1,83 |
| Al2O3 | 10,97 | 11,58 | 12,27 | 11,20 | 11,05 | 11,90 | 13,70 | 10,99 | 11,47 | 11,25 | 12,21 | 12,42 | 14,84 | 10,55 | 14,61 |
| SiO2 | 37,85 | 42,23 | 43,40 | 38,03 | 46,91 | 37,97 | 44,56 | 41,05 | 47,24 | 42,75 | 47,26 | 38,06 | 46,14 | 34,10 | 43,69 |
| K2O | 2,07 | 2,16 | 2,58 | 2,14 | 2,06 | 2,17 | 2,65 | 1,98 | 2,27 | 1,95 | 2,74 | 2,33 | 2,48 | 1,85 | 3,92 |
| CaO | 1,05 | 1,18 | 1,29 | 1,14 | 1,05 | 2,11 | 1,67 | 0,99 | 1,34 | 0,55 | 0,45 | 0,50 | 0,24 | 0,81 | 0,37 |
| TiO2 | 0,33 | 0,49 | 0,31 | 0,39 | 0,36 | 0,29 | 0,47 | 0,41 | 0,31 | 0,34 | 0,39 | 0,29 | 0,64 | 0,36 | 0,42 |
| FeO | 3,06 | 3,46 | 3,24 | 3,34 | 3,23 | 4,02 | 4,57 | 3,37 | 3,41 | 3,31 | 3,60 | 3,71 | 4,77 | 4,32 | 4,40 |
| Sum (wt%) | 58,29 | 65,07 | 66,68 | 59,08 | 67,80 | 61,63 | 71,45 | 61,88 | 69,12 | 63,49 | 69,85 | 60,39 | 72,91 | 54,95 | 70,35 |

**Table 3: Averaged EPMA analyses of major element concentrations of the clay-size fraction within shear zones, veins and the host sediment matrix.**





| Structure Datum ref. | Matrix af_4 | Matrix af_5 | Matrix af_6 | Matrix yc_4 | Matrix yc_5 | Matrix da_6 | Matrix da_8 | Matrix da_9 | Shear zone yc_1 | Shear zone yc_3 | Shear zone yc_7 | Shear zone da_1 | Shear zone da_2 | Shear zone da_4 | Vein af_1 | Vein af_2 | Vein af_3 | Vein af_7 | Vein af_8 | Vein da_5 |
|---|---|---|---|---|---|---|---|---|---|---|---|---|---|---|---|---|---|---|---|---|
| Li | 51,46 | 57,45 | 54,65 | 41,55 | 55,61 | 37,90 | 50,44 | 46,31 | 56,44 | 62,95 | 55,12 | 53,70 | 56,43 | 51,04 | 60,87 | 63,26 | 61,78 | 52,53 | 73,78 | 69,28 |
| B | 89,21 | 91,86 | 88,55 | 88,92 | 68,77 | 55,73 | 70,40 | 66,68 | 83,31 | 80,66 | 80,06 | 69,83 | 77,52 | 89,82 | 105,34 | 103,76 | 103,08 | 83,72 | 140,51 | 115,56 |
| Si* | 18,23 | 18,23 | 18,23 | 18,23 | 18,23 | 18,23 | 18,23 | 18,23 | 22,08 | 22,08 | 22,08 | 22,08 | 22,08 | 22,08 | 21,36 | 21,36 | 21,36 | 21,36 | 21,36 | 21,36 |
| Ca* | 0,35 | 0,42 | 0,41 | 0,56 | 0,49 | 0,94 | 0,72 | 0,98 | 0,93 | 0,90 | 1,15 | 1,01 | 0,71 | 0,63 | 0,46 | 0,45 | 0,42 | 0,62 | 0,80 | 0,87 |
| Sc | 11,42 | 11,43 | 11,43 | 11,15 | 10,93 | 9,16 | 12,73 | 9,58 | 13,63 | 13,92 | 13,34 | 13,43 | 14,17 | 13,12 | 13,16 | 13,92 | 13,04 | 12,33 | 15,12 | 14,24 |
| Ti* | 0,25 | 0,28 | 0,37 | 0,29 | 0,22 | 0,22 | 0,49 | 0,24 | 0,31 | 0,32 | 0,30 | 0,31 | 0,37 | 0,32 | 0,35 | 0,33 | 0,35 | 0,38 | 0,38 | 0,35 |
| V | 92,02 | 91,84 | 84,85 | 82,52 | 82,53 | 68,84 | 91,84 | 78,23 | 101,78 | 104,55 | 99,71 | 99,65 | 102,14 | 101,32 | 102,87 | 109,10 | 104,58 | 91,82 | 127,47 | 115,80 |
| Cr | 49,50 | 52,44 | 46,81 | 48,65 | 44,37 | 38,48 | 50,92 | 41,27 | 57,15 | 58,20 | 59,86 | 58,82 | 63,81 | 135,00 | 69,87 | 63,89 | 59,54 | 43,43 | 71,28 | 68,50 |
| Co | 9,08 | 9,53 | 9,06 | 7,02 | 6,82 | 6,07 | 8,62 | 6,85 | 9,34 | 9,45 | 10,07 | 9,62 | 8,82 | 12,06 | 9,77 | 10,78 | 9,49 | 12,42 | 11,85 | 12,35 |
| Ni | 29,06 | 30,44 | 28,79 | 24,07 | 20,48 | b.d.l. | 32,25 | 17,19 | 29,86 | 29,56 | 30,23 | b.d.l. | b.d.l. | b.d.l. | 37,81 | 37,13 | 32,37 | b.d.l. | 44,74 | 46,69 |
| Cu | 34,07 | 37,42 | 29,39 | 34,94 | 35,82 | 28,96 | 60,30 | 41,68 | 50,67 | 59,29 | 48,46 | 52,48 | 48,94 | 49,74 | 41,43 | 52,85 | 35,04 | 44,83 | 64,92 | 76,91 |
| Zn | 130,89 | 192,86 | 132,07 | 152,78 | 150,12 | 236,31 | 175,31 | 171,65 | 173,02 | 185,60 | 195,42 | 218,34 | 202,89 | 187,62 | 152,84 | 174,20 | 140,29 | 223,22 | 163,55 | 157,12 |
| As | 2,83 | 3,14 | 2,81 | 2,79 | 2,27 | 1,99 | 3,39 | 1,92 | 5,10 | 4,68 | 3,64 | 4,42 | 4,71 | 7,54 | 3,26 | 3,33 | 3,21 | 3,25 | 4,11 | 4,86 |
| Rb | 97,59 | 104,01 | 91,27 | 80,82 | 79,09 | 71,39 | 84,66 | 73,13 | 96,83 | 95,57 | 95,52 | 95,50 | 100,18 | 90,97 | 116,85 | 121,53 | 117,01 | 101,87 | 128,62 | 124,74 |
| Sr | 50,04 | 91,76 | 60,52 | 58,55 | 53,46 | 99,86 | 124,83 | 88,62 | 89,65 | 85,38 | 94,85 | 132,41 | 83,04 | 129,44 | 56,46 | 60,21 | 59,57 | 92,92 | 62,13 | 69,98 |
| Y | 12,58 | 18,09 | 12,73 | 11,32 | 15,51 | 9,14 | 14,41 | 7,83 | 14,04 | 14,14 | 14,05 | 14,40 | 12,10 | 11,85 | 12,79 | 16,08 | 16,64 | 12,72 | 15,34 | 14,07 |
| Zr | 104,91 | 59,33 | 62,93 | 62,22 | 86,12 | 45,89 | 67,02 | 44,10 | 78,23 | 76,30 | 73,12 | 64,98 | 62,90 | 71,36 | 73,30 | 83,92 | 80,65 | 61,02 | 80,42 | 77,83 |
| Nb | 5,58 | 6,70 | 11,73 | 7,11 | 6,48 | 4,61 | 13,59 | 5,90 | 7,68 | 7,96 | 7,56 | 6,94 | 9,30 | 7,95 | 6,83 | 7,28 | 7,56 | 8,39 | 8,17 | 7,86 |
| Sb | 0,92 | 0,81 | 0,89 | 0,83 | 0,75 | 0,58 | 0,89 | 0,70 | 1,01 | 1,14 | 0,99 | 0,93 | 0,90 | 0,97 | 0,96 | 0,93 | 0,99 | 0,82 | 1,40 | 1,11 |
| Cs | 7,23 | 7,92 | 7,09 | 6,19 | 5,73 | 4,81 | 6,23 | 5,39 | 7,15 | 7,28 | 7,03 | 7,22 | 7,54 | 6,45 | 9,32 | 9,37 | 8,94 | 9,84 | 10,13 | 9,80 |
| Ba | 246,55 | 1091,58 | 208,79 | 236,54 | 212,58 | 593,52 | 918,80 | 278,77 | 263,78 | 262,77 | 333,69 | 290,44 | 336,20 | 313,95 | 284,21 | 384,43 | 263,41 | 359,46 | 242,83 | 260,22 |
| La | 17,95 | 15,67 | 15,11 | 11,91 | 11,67 | 12,73 | 18,27 | 11,17 | 18,84 | 20,36 | 16,85 | 18,39 | 19,77 | 16,21 | 22,65 | 20,67 | 19,58 | 20,42 | 17,41 | 17,66 |
| Ce | 32,65 | 34,70 | 30,02 | 25,35 | 24,46 | 25,83 | 42,01 | 25,77 | 40,76 | 46,54 | 37,83 | 38,22 | 40,46 | 33,57 | 48,55 | 47,32 | 44,48 | 47,71 | 38,01 | 40,93 |
| Pr | 3,43 | 3,85 | 3,30 | 2,79 | 2,96 | 2,80 | 4,44 | 2,69 | 4,29 | 4,63 | 3,95 | 3,99 | 4,17 | 3,44 | 4,98 | 4,56 | 4,52 | 5,10 | 3,91 | 4,34 |
| Nd | 13,33 | 16,92 | 12,87 | 10,56 | 11,84 | 10,34 | 17,07 | 9,78 | 16,07 | 17,24 | 14,89 | 14,98 | 15,46 | 12,92 | 18,74 | 17,85 | 17,32 | 19,19 | 15,25 | 16,00 |
| Sm | 2,67 | 3,53 | 2,66 | 2,15 | 2,60 | 1,82 | 3,32 | 1,80 | 2,96 | 3,25 | 2,90 | 2,72 | 2,86 | 2,50 | 3,68 | 3,57 | 3,43 | 3,26 | 3,22 | 3,41 |
| Eu | 0,58 | 0,79 | 0,62 | 0,43 | 0,51 | 0,44 | 0,74 | 0,40 | 0,67 | 0,70 | 0,67 | 0,59 | 0,60 | 0,55 | 0,61 | 0,78 | 0,64 | 0,68 | 0,71 | 0,70 |
| Gd | 2,31 | 3,27 | 2,21 | 1,98 | 2,61 | 1,71 | 3,06 | 1,55 | 2,97 | 3,00 | 2,95 | 2,48 | 2,53 | 2,28 | 2,85 | 3,38 | 2,70 | 2,79 | 2,76 | 2,56 |
| Tb | 0,34 | 0,49 | 0,36 | 0,31 | 0,43 | 0,26 | 0,46 | 0,23 | 0,41 | 0,44 | 0,42 | 0,39 | 0,37 | 0,34 | 0,39 | 0,46 | 0,42 | 0,38 | 0,41 | 0,41 |
| Dy | 2,33 | 3,15 | 2,23 | 2,04 | 2,89 | 1,68 | 2,99 | 1,49 | 2,63 | 2,71 | 2,61 | 2,63 | 2,36 | 2,26 | 2,52 | 2,93 | 2,96 | 2,44 | 2,82 | 2,62 |
| Ho | 0,48 | 0,69 | 0,44 | 0,44 | 0,63 | 0,36 | 0,60 | 0,31 | 0,55 | 0,57 | 0,56 | 0,57 | 0,50 | 0,47 | 0,51 | 0,59 | 0,59 | 0,49 | 0,54 | 0,55 |
| Er | 1,41 | 1,94 | 1,37 | 1,27 | 1,90 | 1,07 | 1,70 | 0,91 | 1,62 | 1,64 | 1,65 | 1,75 | 1,44 | 1,40 | 1,43 | 1,66 | 1,79 | 1,49 | 1,60 | 1,57 |
| Tm | 0,20 | 0,28 | 0,22 | 0,20 | 0,30 | 0,16 | 0,26 | 0,15 | 0,25 | 0,25 | 0,26 | 0,27 | 0,22 | 0,22 | 0,24 | 0,27 | 0,30 | 0,22 | 0,24 | 0,24 |
| Yb | 1,49 | 1,86 | 1,59 | 1,42 | 2,15 | 1,18 | 1,82 | 1,01 | 1,80 | 1,78 | 1,79 | 1,88 | 1,58 | 1,63 | 1,56 | 1,85 | 2,16 | 1,54 | 1,88 | 1,83 |
| Lu | 0,21 | 0,28 | 0,23 | 0,22 | 0,32 | 0,18 | 0,27 | 0,15 | 0,29 | 0,27 | 0,27 | 0,29 | 0,24 | 0,24 | 0,23 | 0,27 | 0,32 | 0,24 | 0,27 | 0,27 |
| Hf | 2,67 | 1,77 | 1,92 | 1,96 | 2,54 | 1,36 | 2,15 | 1,34 | 2,49 | 2,61 | 2,29 | 2,08 | 2,01 | 2,27 | 2,20 | 2,72 | 2,67 | 1,96 | 2,28 | 2,36 |
| Ta | 0,47 | 0,51 | 1,12 | 0,55 | 0,53 | 0,37 | 0,86 | 0,48 | 0,64 | 0,65 | 0,66 | 0,58 | 0,83 | 0,64 | 0,54 | 0,56 | 0,59 | 0,68 | 0,61 | 0,58 |
| Pb | 13,36 | 15,31 | 20,83 | 11,15 | 8,75 | 8,48 | 14,93 | 10,32 | 13,15 | 17,48 | 14,91 | 16,43 | 16,15 | 19,51 | 15,96 | 17,51 | 15,79 | 13,96 | 23,34 | 19,82 |
| Th | 7,03 | 7,78 | 7,50 | 6,21 | 6,58 | 5,17 | 8,40 | 5,14 | 9,01 | 8,74 | 8,20 | 7,73 | 8,77 | 8,10 | 9,74 | 8,91 | 10,13 | 8,72 | 8,62 | 8,12 |
| U | 1,58 | 1,69 | 1,98 | 1,56 | 1,89 | 1,18 | 1,79 | 1,44 | 1,98 | 2,06 | 1,90 | 1,75 | 1,77 | 2,00 | 1,79 | 2,37 | 2,06 | 1,74 | 1,96 | 2,05 |





**Table 4: Trace element concentrations in the sediment matrix, in shear zones, and in veins measured by HC-LA-ICPMS (sample 21R-2, 82-85). All the element concentrations are expressed in ppm except Si\*, Ca\* and Ti\* that are expressed in wt%.**