# Peer review of "Deformation-enhanced diagenesis and bacterial proliferation in the Nankai accretionary prism"

_Solid Earth, 2021_

## Author Response (AR1)

Rebuttal letter, manuscript « **Deformation-enhanced diagenesis and bacterial proliferation in the Nankai accretionary prism" by** Vincent Famin, Hugues Raimbourg, Muriel Andreani, and Anne-Marie Boullier

Please find below our responses (in red) to the reviewers' comments (copied and pasted hereafter, in black).

**Reviewer #1**

1.      My main concern is about the number of samples/analyses presented to arrive at the conclusions listed above. Notably, the inference of the increase in illite crystallinity in deformation bands is based on XRD analysis of one sample and analysis of the trace elements in one other sample. This is very limited evidence. In the absence of more supporting data, I feel that the authors should weaken some of their statements and make it clear that they are speculating.

To address this comment we made the following changes:

-       Abstract, lines 18-22, the text has been modified to stress that the conclusion of smectite to illite transformation is based on two techniques applied on two samples: "In tandem, one shear zone sample displays a destabilization of smectite or illite/smectite mixed layers and a slight crystallization of illite relative to its sediment matrix, and another sample shows correlated increases in B and Li in shear zones and veins compared to the host sediment, both effects suggesting a transformation of smectite into illite in deformation bands. The two diagenetic reactions of sulfide precipitation and smectite to illite transformation are explained by a combined action of sulfate-reducing and methanogen bacteria, […]"
-       Discussion, section 5.3, lines 284-288, the following sentence has been modified to state the exact number of samples and analyses on which our illitization interpretation is based: "In the only sample studied by XRD (4R-3, 73-76), the shear zone displays a disappearance of smectite or I/S mixed layers, and an increased crystallinity of illite, relative to its host matrix. In the other sample studied for trace elements (21R-2, 82-85), the correlated B and Li enrichments of the six shear zone and six vein analyses relative to the matrix, particularly noticeable in veins (Fig. 11), are two additional arguments suggesting that deformation bands localize smectite transformation into illite."
-       At the end of section 5.3, we added the following sentences to admit that our conclusion is still speculative, lines 301-310: "This conclusion is speculative for the time being given the small corpus of data presented here (one shear zone sample analyzed for XRD and twelve trace element analyses of shear zones and veins in another sample). Future work will have to test the reproducibility of these findings and their applicability at larger scale in accretionary prism."

2.      Line 30: please add a reference after "in the accretionary prism".

The following references have been added (line 40):
-       For the mechanics of accretionary prisms: Davis, D., Suppe, J., and Dahlen, F. A.: Mechanics of fold-and-thrust belts and accretionary wedges, *Journal of Geophysical Research*, **88**, 1153–1172, 1983.
-       For the influence of fluid pressure on shallow seismicity: Moore, J. C., and Saffer, D. M.: Updip limit of the seismogenic zone beneath the accretionary prism of southwest Japan:

An effect of diagenetic to low-grade metamorphic processes and increasing effective stress. *Geology*, 29, 183–186, 2001.

3. Line 30-31: please add references for the "large amount of work".

Four already cited reference have been added, lines 42-43 (Brown et al., 2001; Henry and Bourlange, 2004; Pohlmann et al., 2009; Raimbourg et al., 2017), plus one new reference: Kastner, M., Elderfield, H., and Martin, J. B.: Fluids in convergent margins: What do we know about their composition, origin, role in diagenesis and importance for oceanic chemical fluxes?. Philosophical Transactions of the Royal Society A, 335, 243–259, doi:10.1098/rsta.1991.0045, 1991.

4. Line 81-82: please mention that the specifics of the samples studied can be found in Table 1.

We cannot mention the specificities of the samples lines 92-93 because deformation structures are not yet described at this stage of the text. To take this comment into account, we added "(listed in Table 1)" after "The studied samples…" line 92, and we modified the first sentence of the Methods section to mention that the sampled deformation bands and the analyses performed on them are summarized in Table 1 (lines 122-123).

5. Methods section: it would be helpful if sub-sections would be added.

Three subsections have been added: 3.1 X-ray diffraction (line 128); 3.2 Major element maps and quantitative analyses (line 144); 3.3 Trace-element analyses (line 160)

6. Line 112: "Secondary" presumably "Scanning" is meant?

Indeed, now corrected line 125

7. Line 138: please replace "the analysis on" by "the analysis of".

Done, now line 155.

8. Line 153: please define "BIR".

BIR-1 is the full name of this reference rock material from USGS. Having checked in many publications, including studies in Geostandard Newsletter, the initials of this name are never mentioned (they refer to basalt of the Island Ridge). We replaced "BIR" by "BIR-1" in the Methods (line 172-73), in Table A1 and in its caption, and specify that it is run as an unknown (line 173).

9. Line 158: please clarify "those samples".

We removed "those" line 177, and specified that pyrite was found in "all" the samples, line 178, to be consistent with the description of barite, found in only one sample (as said line 191, not modified).

10. Line 165: please clarify "indifferently".

We replaced « indifferently » by « both », line 184.

11.     Line 182: please replace "An example of SEM element map" by "An example of an SEM element map".

Modified by "An example of SEM element maps" (lines 205-206) as there are three maps in Figure 5.

12.     Line 188: "S" this element is not shown in Figs. 7 and 9.

True, "S" is shown in Figure 5, now added line 211.

13.     Line 207-208: "This greater compaction is seen in the SEM and XRF maps." Please explain what observations lead to this statement.

The sentence has been modified as follows to be more explicit (line 230): "This greater compaction is indicated by the general increases in element concentrations observed in SEM and XRF maps."

14.     Line 212: please replace "confirms" by "supports".

Done, line 238.

15.     Line 257: please replace "show" by "suggest".

Done, line 283.

16.     Line 270: "reach" should probably be replaced by "obtain".

We prefer to keep the exact term "reach" used by Esnault (2013) cited in this sentence (line 297).

17.     Line 280: please clarify "their".

We replaced « their concentration » by "the concentration of these compounds", line 315.

18.     Line 295: please replace "show" by "suggest".

Done, line 343.

19.     Line 320: please replace "tiny" by "microscopic".

We replaced "tiny" by "small" as deformation bands are not microscopic, line 354.

20.     Figure 1c: Please clarify the right most part of the figure, notably "Nb". Presumably this is "number", which is usually abbreviated as "Nr.".

"Nb" replaced by "Nr" in Figure 1c, and also in Table 2.

21.    Table 2: the volume percentages are reported down to the third decimal, which seem improbably precise to me. What precision can be expected from the analysis by ImageJ?

The precision of ImageJ depends on the number of pixels in the pictures, as the volume percentage of pyrite is estimated by the total area of bright pixels relative to the total surface of the picture. Our pictures have been taken with a 11 Mpixel camera. The precision is of the fifth or sixth decimal depending on the brightness chosen to represent pyrite. However, we agree with the reviewer that we do not need such a precision, and rounded the numbers to the second decimal in Table 2.

22.    Table 3: where is the data from the other 2 samples that have been analyzed by EPMA according to Table 1?

Indeed, one sample (10R-2, 2-12, thin section VFC15) was missing in Table 3. The other sample (21R-2, 82-85) includes in fact two thin sections (VFC3 and VFC4). The new Table 3 has been corrected to include the missing sample and to show the thin sections.

23.    Table 4: where are the results of the other sample that was analysed by HC-LA-ICPMS according to Table 1?

The two slabs (VFC3 and VFC4) coming from the same sample (21R-2, 82-85) were analyzed. All the results from these two slabs are shown in Table 4. The caption has been modified to explain that the two slabs have been analyzed. Line 162, we also specified that the two slabs VFC3 and VFC4 were used for trace element analyses.

**Reviewer #2**

Specific comments

The finding presented are quite compelling. The changes in chemistry and mineralogy between mm-scale deformation structures and matrix are suggestive of fundamental processes that seem to link biology and strain localization. The number of samples that have provided the key findings is very limited, reflecting the exacting work and the very small supply of core material.  A broader discussion of the context would highlight additional implications that might be worthwhile to motivate work aimed at reproducing the findings and scaling up the applicability.  A couple are presented here for consideration.

Site C0001 is in the footwall of a megasplay fault that is crosscut by a slope basin or mass-transport deposit.  It is also in the hangingwall of other out-of-sequence thrust faults. The structural setting of the 3.5-5.5 Ma sediments examined could be mechanically connected to either or both of these structures, but the constraints provided in the manuscript suggest that the structures were formed prior to the emplacement of these faults. Is the timing well constrained? If the dewatering is exclusively burial-related (consistent with the fact that the shear zones record normal motion), the link to more mature faults is more of a stretch. Is it possible that the structures are synchronous with the nearby out-of-sequence thrusts?

We thank the reviewer for this very important and useful comment. Indeed, we do have temporal constraints on the formation of deformation bands. We first removed the faults from the histogram log of deformation structures in Figure 1b, because Lewis et al. (2013) showed that many of them were induced by drilling. The modified histogram shows that the deformation bands we are studying (shear zones and veins) almost exclusively occur in the accretionary prism and not in the slope apron. The consequence is that the deformation bands cannot be related to burial (otherwise they would be found in the two units), and must be of tectonic origin. The deformations bands thus formed during accretion and before the deposition of the slope sediment. This is also the timing of activity of the megasplay thrust fault uphill of site C0001. We can thus conclude that the microstructures and the major thrust roughly correspond to the same time intervalley. The manuscript was modified as follows:

- Lines 104-105, we mention that "This study hereafter focuses on shear zones due to their larger thickness than faults, and because Lewis et al. (2013) showed that many of the faults are in fact drilling-induced."
- We modified Figure 1c to show only shear zones and veins.
- We added a new paragraph lines 323-333: "Another important question concerns the timing of deformation bands and their bacterial proliferation. Given the need of nutriments for metabolic reactions, it is tempting to interpret these structures as formed at shallow depth below the sea floor, in proximity of seawater sulfate supply. However, shear zones as well as veins were almost exclusively found in the accretionary prism (Unit II) and not in the slope sediment (Unit I) above the unconformity (Fig. 1c). This fact implies that most of the deformation bands studied here are not burial-related, but are rather associated with the tectonics of the accretionary prism. A way to reconcile the two inferences is to suggest that deformation bands, and biological diagenesis in them, developed in the upper portion of the accretionary prism during thrusting, and before the deposition of slope sediments. Whether deformation bands are mechanically compatible with thrusting is unfortunately unknown because no kinematics could be assigned to the majority of them. Nevertheless, we note that this proposed timing coincides with the activity of the megasplay fault thrust uphill of C0001 (Fig. 1b). It is thus possible that deformation bands may represent early stages of strain localization, and fluid expulsion, in the context of megasplay fault development."

These overarching questions connect to the implications that start on Line 328, specifically the potential relation between structures of the kind the authors nicely characterize and larger faults. The timing questions are thus critical. Given how much interest there is on accretionary prisms, and on the spectrum of time scales over which seismic energy is released, the following specific questions might be worth consideration. For example, if the dewatering structures are tectonic (not simply products of burial) over what timeframes can the authors bracket them to have been active? Are they "one-time" features related to a single seismic event on the megathrust or the more proximal megasplays? Alternatively, are they formed over many seismic cycles, or perhaps even during the inter-seismic phase as fault zone coupling evolves? Are they crosscut by structures with known kinematics that help narrow the timing? Some of the structures are themselves normal sense shear zones. I realize that these questions are challenging. I ask because I wonder whether the fluids produced by the processes described might have been supplied to structures up dip, down dip or even along strike? If these structures immediately precede the development of throughgoing faults, do they shed light on process-zone evolution? If they are coeval with throughgoing faults, do they help us understand feedbacks between damage zones and faults? An entirely different implication/question: could these structures be signatures of aseismic processes such as tremor?

These questions are very interesting, unfortunately we cannot answer them. We do not have a more precise timing for the development of deformation bands, because no kinematics could be assigned to the majority of them, and because core observation seldom provides information about crosscutting relationships. We do not know whether these structures are "one time events", whether they are caused by earthquakes or slow-slip events or anything else. Given our lack of knowledge, we do not feel confident at discussing these points in this manuscript. Nevertheless, we added a sentence to suggest the possibility that microstructures might supply fluids to larger faults, given their coeval timing, lines 348-350: "Given the temporal consistency between megasplay faulting and deformation bands, the dewatering of these many microstructures could be supplied to major faults, which might explain some freshwater fluxes observed in accretionary prisms (e.g. Kastner et al., 1993; Vrolijk et al., 1991)."

These questions require timing constraints that are very difficult, if not impossible. Given that the "plumbing of accretionary prisms" remains a hot topic, as the authors rightly note, providing the larger context would elevate the significance of the findings, motivating additional innovative work of this sort. In the Discussion section, the authors agree with prior work that these deformation structures are in effect byproducts of dewatering. Is it possible that volume fluxes can be estimated so that we have a better sense of the scale? Getting word out about the findings strikes me as important because as more core from similar settings is collected, the community might prioritize this kind of work so that we can begin evaluating the true scales over which these processes operate. The potential feedbacks between biogenic and tectonic processes are quite provocative. If the structures are well constrained to pre-date the megasplays, then the implications for fault evolution are more tenuous and my many questions are not so helpful.

We can neither estimate the fluxes of dewatering produced by microstructures because we do not have a quantified budget of freshwater consumed or produced by bacterial proliferation. This question will require a large amount of experimental work to be solved. We already mentioned in the previous version that quantitative work is needed to assess the importance of biogenic diagenesis in the fluid budget (lines 350-351). In our opinion, our interpretation stretches already very far in the implications section, and we prefer not to discuss this point in more detail.

Technical corrections

Line 157: "Petrographic" might be more appropriate than "Petrologic" here given that much of what follows is derived from thin-section work it seems.

Corrected, now lines 176-177.

Line 265: The topic has received "much attention." This might be crisper than "a large attention…"

Corrected, now line 291.

Line 339: The sentence that starts "This dual biogenic..." is confusing.

We removed this sentence and replaced it by the following sentence, lines 391-392: "Both diagenetic reactions occurred during the development of deformation bands and vanished afterwards. This biogenic diagenesis may be explained by a locally enhanced activity of anaerobic microorganism in deformation bands, which may be related to the generation of $H_2$ by intracrystalline deformation of silicates minerals."